# KDSalBox: A toolbox of efficient knowledge-distilled saliency models

**Ard Kastrati**[1,2*]
akastrati@ethz.ch

**Zoya Bylinskii**[2]
bylinski@adobe.com

**Eli Shechtman**[2]
elishe@adobe.com

[1]ETH Zurich, Switzerland
[2]Adobe Research, US

## Abstract

Dozens of saliency models have been designed over the last few decades, targeted at diverse applications ranging from image compression and retargeting to robot navigation, surveillance, and distractor detection. Barriers to their use include the different and often incompatible software environments that they rely on, as well as the computational inefficiency of older implementations. For application-purposes models are then frequently chosen based on convenience and efficiency, at the expense of optimizing for task performance. To facilitate the evaluation and selection of saliency models for different applications, we present KDSalBox - a toolbox of 10 knowledge-distilled saliency models. Using the original model implementations available in their native environments, we produce saliency training data for efficient MobileNet-based architectures, that are identical in their architecture but differ in how they learn to distribute saliency over an image. The resulting toolbox allows these 10 models to be efficiently run, compared, and be practically applied.

## 1 Introduction

At the speed at which technology and software libraries change, image processing applications like segmentation, stylization, etc., become quickly outdated. New implementations replace older ones, in some cases re-inventing prior solutions, and in others supplanting older algorithms with newer, data-driven ones. In the case of saliency modeling, the top performing models on saliency benchmarks (Kümmerer et al. (2019); Yu et al. (2017)) are deep neural networks, all trained on the same dataset of visual attention data - the only one of its size (Jiang et al. (2015)). As a result, the top models all fall prey to similar biases and have similar limitations (Bruce et al. (2015); Bylinskii et al. (2016)). At the same time, older models including the one by Itti et al. (1998b), based on low-level image features rather than being directly trained on eye movement data fare better for certain applications (Bruce et al. (2015)). For these reasons, they continue to serve as comparisons in saliency papers today.

On the other hand, the increasing availability of constrained computing environments (e.g., mobile phones) have led to increased focus on the design of efficient models. A number of compact architectures have been proposed recently (Sandler et al. (2018); Tan & Le (2019)). These models are carefully designed with components such as inverted bottleneck units, depthwise separability and pointwise convolutions with batch normalization. Key techniques that have been used to train these networks include model compression and knowledge distillation (Caruana et al. (2006); Hinton et al. (2015)), where a student (the efficient network) uses the teacher's (the complex network) supervision during the training process. Knowledge distillation of neural networks commonly uses

---

[*]This work was completed as part of an Adobe Research internship.

3rd Workshop on Shared Visual Representations in Human and Machine Intelligence (SVRHM 2021) of the Neural Information Processing Systems (NeurIPS) conference, Virtual.

the intermediate activations as hints (Romero et al. (2014)) for better transfer learning. However, the nature of the student-teacher paradigm allows to use the teacher in a black-box manner where the cross-entropy of only the output of the teacher and the student is backpropagated as a signal to update the weights (Hinton et al. (2015)). Recent work in saliency has shown that efficient models for saliency prediction can perform on par with the more complex models while being several orders of magnitude faster (Hu & McGuinness (2020); Zhang et al. (2019)).

The sheer quantity and diversity of the available computational saliency models makes it difficult to evaluate what ideas or mechanisms are generalizable, scalable or better suited for a given application. To this end, the MIT/Tuebingen Saliency Benchmark (Kümmerer et al. (2019)) was introduced and has shown to be beneficial for driving progress in saliency research. The MIT/Tuebingen Saliency Benchmark provides for each model only a coarse summary of performance and a model ranking based on a set of standardized metrics. However, beyond comparing summary numbers, re-implementing or re-running these models to evaluate them on new datasets and applications often proves difficult due to different (and occasionally deprecated or outdated) technical requirements, environments, and dependencies. A few saliency frameworks (Kuemmerer et al. (2015), Wloka et al. (2018)) have recently been proposed to reduce this effort and create a standardized interface to many saliency models at once. These frameworks not only help researchers explore these models but also facilitate reproducibility. While these frameworks can facilitate saliency research, continuous maintenance of these frameworks and dependencies is required as the number of models continues to grow. Wloka et al. (2018) tackles this issue by providing containerized images for each model. However, practical technical barriers remain, such as that all Matlab-based models require a licence and are typically not efficient to run. Given that saliency is often used as a preprocessing step for downstream image processing tasks, the traditional (non-deep) models are often rendered impractical, even though the predictions they make may be better suited for some applications. Here we present a technical report to supplement the release of our KDSalBox toolbox of saliency models[2]. Our toolbox is available to the research community to facilitate the evaluation and use of saliency models in different applications, by removing technical barriers. Our toolbox provides a one-stop shop for different saliency models, equivalent in dependencies and compute time.

## 2 The KDSalBox

Our toolbox consists of ten knowledge-distilled saliency models that try to mimic as closely as possible their original implementations. This approach provides a unified framework for saliency research that is easy to maintain, facilitates the exploration, evaluation and selection of saliency models, and levels the playing field for different types of models (e.g., including Matlab-based libraries with bulky wrappers). All the knowledged-distilled models are identical in their architecture but only differ in how they learn to distribute saliency over an image. This makes the framework easy to use, and we show later that despite its simplicity, the knowledge-distilled models learn to behave very closely to the original counterparts.

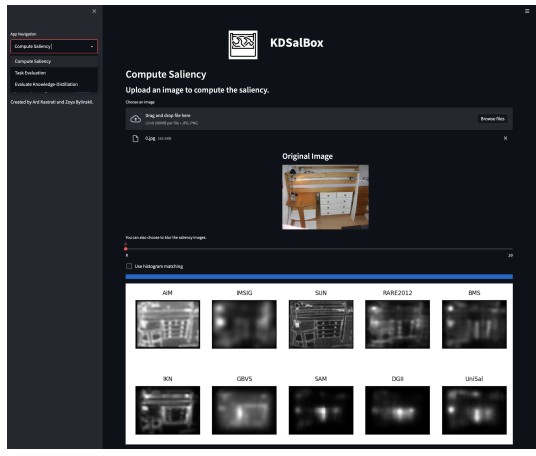

Figure 1: A screenshot of the Streamlit user interface for accessing the KDSalBox.

**Interface of KDSalBox.** Our toolbox offers a similar interface as Wloka et al. (2018). This includes pre- and postprocessing steps such as center prior, blurring, scaling as well as standardized evaluation (Bylinskii et al. (2018)). In addition, due to the simplicity and unified design of the framework, it also provides the infrastructure to distill the knowledge of new models with minimal effort. Finally, we also implemented a graphical user interface to plot the saliency maps (see Figure 1) with the goal to facilitate the evaluation and selection of saliency models for different applications.

---

[2]Available at: `https://github.com/ardkastrati/kdsalbox`

**Ten saliency models.** The models that the toolbox currently includes are the following: AIM (Bruce & Tsotsos (2009)), IKN (Itti et al. (1998a)), GBVS (Harel et al. (2007)), BMS (Zhang & Sclaroff (2013)), IMSIG (Hou et al. (2012)), RARE2012 (Riche et al. (2013)), SUN (Zhang et al. (2008)), UniSal (Droste et al. (2020)), SAM (Cornia et al. (2018)), DGII (Kümmerer et al. (2016)). The first seven models are originally implemented in Matlab and mostly focus on lower level image features. The remaining three are deep neural networks that capture higher level object-based features (such as understanding of objects, faces, etc.) and perform the best on current saliency benchmarks, where they are tested on predicting eye movements on natural images (Kümmerer et al. (2019)).

## 2.1 Knowledge distillation

To perform knowledge distillation, we generated saliency maps from each original model on the SALICON dataset (Jiang et al. (2015)). The generated saliency maps were then used for supervision during the training procedure. Note that, we used *only* the generated saliency images, and not the ground truth (eye movements) of the SALICON dataset, as input during training, since our goal was to mimic as closely as possible the original model behaviors, irrespective of similarity to eye movements. The goal is not to train a new model to perform better at the task of eye movement prediction since the target saliency applications may be different.

**Model architecture.** Our model design follows the encoder-decoder pattern. We used the MobileNetV2 (Sandler et al. (2018)) for the encoder and a simple convolutional neural network (composed of seven convolutional layers with batch normalization) as a decoder. The MobileNet-based encoder serves as a lightweight backbone for image feature computation, while the decoder generates the final saliency map. Through some pilot testing, we converged on a decoder that was simple and compact while performing well in knowledge distilling the ten different saliency models. It contains 6 convolutional layers and 5 intermediate bilinear upsampling layers, motivated by the fact that the original MobileNetv2 downsamples the original image (h x w) to a size (h//32 x w//32).

**Training procedure.** For our training and evaluation we used the generated saliency images of each model on the SALICON training and evaluation datasets, respectively. First the encoder was initialized with the pretrained weights on ImageNet. Then for the first three epochs we froze the encoder, and trained the decoder with our generated images. After three epochs the encoder was unfrozen and the training proceeded normally. We used a learning rate of 0.1. During training we used a binary cross entropy loss and the Adam optimizer. We also experimented with other custom losses (such as a weighted combination of KL-divergence, Correlation Coefficient, etc.) but we didn't observe a noticeable improvement. All our models were trained on a TITAN Xp (12G) GPU.

## 3 Evaluation

**Metrics.** Following Bylinskii et al. (2018), we report the distribution-based metrics CC and KL for evaluating saliency models. We use these metrics and not others like NSS and AUC, because the goal is to compare pairs of distributions to each other (student and teacher saliency maps), rather than evaluating against ground truth fixation locations. **CC** measures Pearson's Correlation Coefficient between two saliency maps as a ratio of their covariance to the product of their variances. **KL** is the Kullback-Leibler divergence between two saliency maps, evaluating the loss of information when one of the maps is used to approximate the other. From their definitions, CC is a symmetric metric, while KL is not. Two maps are similar as CC approaches 1 (higher correlation) and KL approaches 0 (lower divergence).

**How well have the students learned?** We evaluate the ability of each of the knowledge distilled student models to approximate their teachers - the original saliency models run in their native code bases. We report the similarity between the saliency maps produced by the teacher and student models on the SALICON validation data of 5000 images, in Table 1. The trained models perform very similarly compared to their original counterparts. Interestingly, the performance of the Matlab-based models was easier to reproduce compared to the deep learning models. We hypothesize that this is due to the limited dataset size during training making it difficult to capture high-level features, as well as the simple model architecture we used.

Table 1: The ten models currently part of the KDSalBox. We include scores comparing the knowledge-distilled student models to the original saliency model implementations. We report the average inference time by using the SMILER (Wloka et al. (2018)) framework and 640×480 sample images. We note that in comparison to the original implementations, all our resulting student models are 41.2 MB in size with an average inference time of 0.4 sec on CPU and 0.01 sec on GPU.

| Model | Original implementation | | | (Dis)similarity between student & original models | | | |
|---|---|---|---|---|---|---|---|
| | | | | Natural Images (SALICON) | | Patterns (CAT2000) | |
| | Codebase | Size | Time (CPU) | CC ↑ | KL ↓ | CC ↑ | KL ↓ |
| AIM | Matlab | 24MB | 10sec | 0.99 ±0.00 | 0.00 ±0.00 | 0.96 ±0.04 | 0.18 ±0.23 |
| IKN | Matlab | 1MB | 6.0sec | 0.97 ±0.01 | 0.02 ±0.01 | 0.78 ±0.09 | 0.14 ±0.06 |
| GBVS | Matlab | 1MB | 6.1sec | 0.98 ±0.01 | 0.01 ±0.01 | 0.75 ±0.09 | 0.17 ±0.07 |
| BMS | Matlab | 5MB | 0.4sec | 0.89 ±0.07 | 0.05 ±0.02 | 0.78 ±0.11 | 0.30 ±0.26 |
| IMSIG | Matlab | 20KB | 5.7sec | 0.96 ±0.04 | 0.02 ±0.02 | 0.71 ±0.15 | 0.16 ±0.13 |
| RARE2012 | Matlab | 20KB | 6.3sec | 0.87 ±0.10 | 0.10 ±0.06 | 0.76 ±0.12 | 0.30 ±0.16 |
| SUN | Matlab | 1MB | 12sec | 0.94 ±0.04 | 0.02 ±0.01 | 0.95 ±0.02 | 0.21 ±0.25 |
| UniSal | Python | 30MB | 0.4sec | 0.92 ±0.05 | 0.11 ±0.06 | 0.68 ±0.14 | 0.46 ±0.36 |
| SAM | Python | 561MB | 7.3sec | 0.82 ±0.13 | 0.28 ±0.17 | 0.48 ±0.15 | 0.87 ±0.40 |
| DGII | Python | 330MB | 4.8sec | 0.90 ±0.08 | 0.14 ±0.09 | 0.68 ±0.16 | 0.48 ±0.42 |

Table 2: Generalizability experiments. Knowledge distilled saliency models trained on natural images from SALICON evaluated on their similarity to the original model implementations on different image types from the CAT2000 and Imp1K saliency datasets.

| Dataset | Metric | AIM | SUN | IKN | GBVS | BMS | RARE2012 | IMSIG | DGII | UniSal | SAM |
|---|---|---|---|---|---|---|---|---|---|---|---|
| **CAT2000** | | | | | | | | | | | |
| Action | CC ↑ | 0.98 ±0.01 | 0.96 ±0.02 | 0.86 ±0.04 | 0.81 ±0.07 | 0.80 ±0.07 | 0.77 ±0.09 | 0.75 ±0.08 | 0.50 ±0.14 | 0.54 ±0.13 | 0.41 ±0.14 |
| | KL ↓ | 0.04 ±0.06 | 0.07 ±0.03 | 0.10 ±0.03 | 0.13 ±0.04 | 0.16 ±0.05 | 0.21 ±0.06 | 0.12 ±0.04 | 1.00 ±0.37 | 0.85 ±0.30 | 1.26 ±0.39 |
| Affective | CC ↑ | 0.97 ±0.01 | 0.96 ±0.02 | 0.85 ±0.04 | 0.81 ±0.07 | 0.80 ±0.08 | 0.77 ±0.10 | 0.74 ±0.11 | 0.50 ±0.15 | 0.59 ±0.13 | 0.49 ±0.13 |
| | KL ↓ | 0.06 ±0.08 | 0.08 ±0.04 | 0.09 ±0.03 | 0.13 ±0.04 | 0.17 ±0.08 | 0.21 ±0.07 | 0.13 ±0.07 | 0.96 ±0.39 | 0.73 ±0.32 | 1.01 ±0.36 |
| Art | CC ↑ | 0.97 ±0.02 | 0.95 ±0.03 | 0.85 ±0.05 | 0.83 ±0.06 | 0.79 ±0.12 | 0.75 ±0.11 | 0.75 ±0.11 | 0.65 ±0.14 | 0.71 ±0.12 | 0.53 ±0.14 |
| | KL ↓ | 0.05 ±0.08 | 0.07 ±0.05 | 0.09 ±0.03 | 0.12 ±0.05 | 0.17 ±0.11 | 0.21 ±0.10 | 0.12 ±0.08 | 0.50 ±0.32 | 0.35 ±0.20 | 0.70 ±0.31 |
| BlackWhite | CC ↑ | 0.96 ±0.02 | 0.94 ±0.02 | 0.86 ±0.04 | 0.86 ±0.05 | 0.80 ±0.06 | 0.74 ±0.06 | 0.55 ±0.12 | 0.55 ±0.14 | 0.67 ±0.12 | 0.49 ±0.13 |
| | KL ↓ | 0.07 ±0.04 | 0.15 ±0.07 | 0.10 ±0.03 | 0.10 ±0.04 | 0.21 ±0.10 | 0.23 ±0.07 | 0.16 ±0.08 | 0.76 ±0.40 | 0.47 ±0.28 | 0.85 ±0.37 |
| Cartoon | CC ↑ | 0.97 ±0.02 | 0.94 ±0.04 | 0.81 ±0.05 | 0.80 ±0.07 | 0.77 ±0.10 | 0.74 ±0.10 | 0.73 ±0.12 | 0.62 ±0.14 | 0.66 ±0.11 | 0.51 ±0.13 |
| | KL ↓ | 0.05 ±0.06 | 0.07 ±0.04 | 0.11 ±0.03 | 0.13 ±0.05 | 0.20 ±0.11 | 0.21 ±0.10 | 0.12 ±0.05 | 0.50 ±0.22 | 0.37 ±0.14 | 0.67 ±0.25 |
| Fractal | CC ↑ | 0.98 ±0.01 | 0.96 ±0.02 | 0.82 ±0.06 | 0.81 ±0.08 | 0.80 ±0.11 | 0.72 ±0.15 | 0.74 ±0.13 | 0.65 ±0.11 | 0.71 ±0.10 | 0.50 ±0.13 |
| | KL ↓ | 0.04 ±0.06 | 0.06 ±0.03 | 0.11 ±0.03 | 0.11 ±0.04 | 0.13 ±0.06 | 0.20 ±0.07 | 0.10 ±0.03 | 0.39 ±0.22 | 0.26 ±0.11 | 0.64 ±0.26 |
| Indoor | CC ↑ | 0.98 ±0.01 | 0.96 ±0.02 | 0.85 ±0.03 | 0.82 ±0.05 | 0.79 ±0.07 | 0.75 ±0.08 | 0.74 ±0.07 | 0.63 ±0.12 | 0.69 ±0.10 | 0.53 ±0.12 |
| | KL ↓ | 0.03 ±0.05 | 0.06 ±0.02 | 0.10 ±0.04 | 0.10 ±0.03 | 0.15 ±0.05 | 0.19 ±0.06 | 0.11 ±0.04 | 0.51 ±0.24 | 0.32 ±0.13 | 0.61 ±0.20 |
| Inverted | CC ↑ | 0.98 ±0.01 | 0.96 ±0.02 | 0.85 ±0.04 | 0.82 ±0.07 | 0.78 ±0.08 | 0.76 ±0.08 | 0.75 ±0.09 | 0.63 ±0.12 | 0.70 ±0.09 | 0.54 ±0.12 |
| | KL ↓ | 0.03 ±0.05 | 0.06 ±0.02 | 0.10 ±0.03 | 0.11 ±0.04 | 0.16 ±0.06 | 0.19 ±0.08 | 0.11 ±0.04 | 0.47 ±0.22 | 0.31 ±0.12 | 0.61 ±0.22 |
| Jumbled | CC ↑ | 0.98 ±0.01 | 0.97 ±0.01 | 0.85 ±0.03 | 0.82 ±0.05 | 0.80 ±0.05 | 0.80 ±0.05 | 0.76 ±0.07 | 0.66 ±0.10 | 0.73 ±0.07 | 0.51 ±0.10 |
| | KL ↓ | 0.07 ±0.08 | 0.08 ±0.03 | 0.09 ±0.03 | 0.12 ±0.04 | 0.17 ±0.06 | 0.19 ±0.06 | 0.10 ±0.05 | 0.42 ±0.21 | 0.26 ±0.08 | 0.62 ±0.18 |
| LineDrawing | CC ↑ | 0.96 ±0.01 | 0.89 ±0.03 | 0.81 ±0.03 | 0.84 ±0.04 | 0.73 ±0.07 | 0.69 ±0.10 | 0.81 ±0.04 | 0.69 ±0.09 | 0.68 ±0.08 | 0.47 ±0.11 |
| | KL ↓ | 0.16 ±0.10 | 0.29 ±0.11 | 0.11 ±0.02 | 0.09 ±0.02 | 0.27 ±0.09 | 0.33 ±0.06 | 0.07 ±0.04 | 0.35 ±0.20 | 0.33 ±0.18 | 0.68 ±0.20 |
| LowResolution | CC ↑ | 0.87 ±0.05 | 0.96 ±0.01 | 0.88 ±0.04 | 0.86 ±0.03 | 0.77 ±0.08 | 0.71 ±0.13 | 0.46 ±0.15 | 0.36 ±0.17 | 0.61 ±0.14 | 0.33 ±0.17 |
| | KL ↓ | 0.08 ±0.03 | 0.08 ±0.02 | 0.09 ±0.04 | 0.08 ±0.02 | 0.22 ±0.07 | 0.20 ±0.10 | 0.18 ±0.07 | 0.83 ±0.30 | 0.43 ±0.17 | 1.02 ±0.36 |
| Noisy | CC ↑ | 0.99 ±0.00 | 0.97 ±0.02 | 0.84 ±0.05 | 0.86 ±0.06 | 0.77 ±0.12 | 0.67 ±0.13 | 0.71 ±0.12 | 0.55 ±0.11 | 0.64 ±0.13 | 0.53 ±0.12 |
| | KL ↓ | 0.03 ±0.05 | 0.04 ±0.01 | 0.10 ±0.04 | 0.08 ±0.03 | 0.10 ±0.04 | 0.15 ±0.05 | 0.09 ±0.03 | 0.55 ±0.21 | 0.39 ±0.21 | 0.62 ±0.23 |
| Object | CC ↑ | 0.96 ±0.02 | 0.96 ±0.02 | 0.85 ±0.05 | 0.82 ±0.06 | 0.80 ±0.10 | 0.80 ±0.10 | 0.75 ±0.11 | 0.61 ±0.15 | 0.69 ±0.12 | 0.53 ±0.14 |
| | KL ↓ | 0.14 ±0.13 | 0.13 ±0.10 | 0.10 ±0.03 | 0.14 ±0.05 | 0.25 ±0.15 | 0.26 ±0.13 | 0.16 ±0.11 | 0.75 ±0.38 | 0.50 ±0.24 | 0.87 ±0.32 |
| OutdoorManMade | CC ↑ | 0.98 ±0.01 | 0.96 ±0.02 | 0.84 ±0.05 | 0.81 ±0.07 | 0.79 ±0.08 | 0.74 ±0.09 | 0.75 ±0.09 | 0.62 ±0.12 | 0.71 ±0.08 | 0.54 ±0.13 |
| | KL ↓ | 0.03 ±0.05 | 0.06 ±0.02 | 0.09 ±0.02 | 0.11 ±0.04 | 0.14 ±0.05 | 0.19 ±0.05 | 0.10 ±0.04 | 0.48 ±0.22 | 0.31 ±0.12 | 0.61 ±0.23 |
| OutdoorNatural | CC ↑ | 0.98 ±0.01 | 0.95 ±0.03 | 0.83 ±0.05 | 0.80 ±0.06 | 0.76 ±0.10 | 0.71 ±0.11 | 0.74 ±0.09 | 0.61 ±0.14 | 0.67 ±0.13 | 0.49 ±0.15 |
| | KL ↓ | 0.02 ±0.03 | 0.05 ±0.02 | 0.10 ±0.03 | 0.11 ±0.03 | 0.17 ±0.05 | 0.17 ±0.05 | 0.10 ±0.04 | 0.48 ±0.30 | 0.38 ±0.27 | 0.71 ±0.29 |
| Pattern | CC ↑ | 0.96 ±0.04 | 0.96 ±0.02 | 0.78 ±0.09 | 0.75 ±0.09 | 0.78 ±0.11 | 0.76 ±0.12 | 0.71 ±0.15 | 0.68 ±0.16 | 0.69 ±0.14 | 0.48 ±0.15 |
| | KL ↓ | 0.18 ±0.23 | 0.21 ±0.25 | 0.14 ±0.06 | 0.17 ±0.07 | 0.30 ±0.26 | 0.30 ±0.16 | 0.15 ±0.12 | 0.48 ±0.42 | 0.46 ±0.36 | 0.87 ±0.40 |
| Random | CC ↑ | 0.93 ±0.04 | 0.97 ±0.01 | 0.88 ±0.03 | 0.83 ±0.06 | 0.82 ±0.07 | 0.79 ±0.06 | 0.75 ±0.08 | 0.63 ±0.11 | 0.73 ±0.08 | 0.51 ±0.14 |
| | KL ↓ | 0.18 ±0.09 | 0.10 ±0.03 | 0.10 ±0.05 | 0.16 ±0.05 | 0.26 ±0.09 | 0.26 ±0.09 | 0.19 ±0.09 | 0.74 ±0.30 | 0.44 ±0.20 | 0.99 ±0.35 |
| Satelite | CC ↑ | 0.98 ±0.01 | 0.94 ±0.04 | 0.79 ±0.05 | 0.80 ±0.07 | 0.67 ±0.14 | 0.61 ±0.15 | 0.68 ±0.12 | 0.64 ±0.10 | 0.64 ±0.12 | 0.44 ±0.13 |
| | KL ↓ | 0.01 ±0.04 | 0.04 ±0.02 | 0.10 ±0.02 | 0.11 ±0.04 | 0.11 ±0.04 | 0.21 ±0.06 | 0.08 ±0.04 | 0.36 ±0.17 | 0.34 ±0.16 | 0.69 ±0.24 |
| Sketch | CC ↑ | 0.92 ±0.02 | 0.93 ±0.02 | 0.80 ±0.03 | 0.77 ±0.08 | 0.78 ±0.05 | 0.74 ±0.04 | 0.81 ±0.04 | 0.76 ±0.09 | 0.73 ±0.07 | 0.59 ±0.09 |
| | KL ↓ | 0.57 ±0.10 | 0.52 ±0.12 | 0.13 ±0.02 | 0.22 ±0.10 | 0.60 ±0.14 | 0.39 ±0.05 | 0.25 ±0.09 | 0.66 ±0.23 | 0.56 ±0.15 | 0.87 ±0.21 |
| Social | CC ↑ | 0.98 ±0.01 | 0.95 ±0.02 | 0.84 ±0.03 | 0.81 ±0.05 | 0.78 ±0.07 | 0.75 ±0.08 | 0.71 ±0.09 | 0.45 ±0.12 | 0.52 ±0.13 | 0.41 ±0.10 |
| | KL ↓ | 0.02 ±0.04 | 0.06 ±0.02 | 0.09 ±0.02 | 0.11 ±0.03 | 0.14 ±0.04 | 0.20 ±0.05 | 0.11 ±0.03 | 1.02 ±0.36 | 0.75 ±0.30 | 1.06 ±0.30 |
| **UMSI** | | | | | | | | | | | |
| Ads | CC ↑ | 0.93 ±0.04 | 0.72 ±0.13 | 0.84 ±0.07 | 0.91 ±0.07 | 0.77 ±0.12 | 0.75 ±0.16 | 0.85 ±0.11 | 0.72 ±0.14 | 0.78 ±0.12 | 0.65 ±0.18 |
| | KL ↓ | 0.02 ±0.04 | 0.23 ±0.14 | 0.10 ±0.04 | 0.05 ±0.04 | 0.13 ±0.07 | 0.19 ±0.10 | 0.06 ±0.04 | 0.35 ±0.21 | 0.23 ±0.15 | 0.50 ±0.25 |
| Infographics | CC ↑ | 0.89 ±0.07 | 0.64 ±0.17 | 0.76 ±0.12 | 0.80 ±0.11 | 0.65 ±0.14 | 0.61 ±0.18 | 0.76 ±0.16 | 0.64 ±0.11 | 0.61 ±0.14 | 0.40 ±0.16 |
| | KL ↓ | 0.04 ±0.05 | 0.38 ±0.21 | 0.12 ±0.05 | 0.10 ±0.06 | 0.22 ±0.11 | 0.31 ±0.15 | 0.10 ±0.03 | 0.39 ±0.15 | 0.38 ±0.19 | 0.73 ±0.22 |
| MobileUIs | CC ↑ | 0.97 ±0.01 | 0.94 ±0.02 | 0.77 ±0.07 | 0.74 ±0.11 | 0.67 ±0.12 | 0.70 ±0.12 | 0.78 ±0.11 | 0.67 ±0.10 | 0.60 ±0.14 | 0.44 ±0.15 |
| | KL ↓ | 0.07 ±0.09 | 0.09 ±0.04 | 0.12 ±0.03 | 0.15 ±0.07 | 0.30 ±0.15 | 0.34 ±0.16 | 0.12 ±0.08 | 0.50 ±0.18 | 0.53 ±0.22 | 0.85 ±0.25 |
| MoviePosters | CC ↑ | 0.93 ±0.04 | 0.76 ±0.12 | 0.84 ±0.06 | 0.88 ±0.09 | 0.78 ±0.09 | 0.75 ±0.13 | 0.85 ±0.08 | 0.68 ±0.14 | 0.73 ±0.15 | 0.58 ±0.18 |
| | KL ↓ | 0.02 ±0.03 | 0.16 ±0.10 | 0.10 ±0.03 | 0.07 ±0.06 | 0.13 ±0.07 | 0.20 ±0.09 | 0.07 ±0.04 | 0.43 ±0.21 | 0.31 ±0.19 | 0.59 ±0.26 |
| Webpages | CC ↑ | 0.96 ±0.01 | 0.83 ±0.04 | 0.86 ±0.04 | 0.85 ±0.06 | 0.76 ±0.08 | 0.78 ±0.11 | 0.77 ±0.09 | 0.66 ±0.11 | 0.66 ±0.09 | 0.52 ±0.14 |
| | KL ↓ | 0.04 ±0.06 | 0.22 ±0.06 | 0.08 ±0.02 | 0.09 ±0.04 | 0.21 ±0.09 | 0.22 ±0.11 | 0.11 ±0.04 | 0.39 ±0.14 | 0.35 ±0.09 | 0.68 ±0.21 |

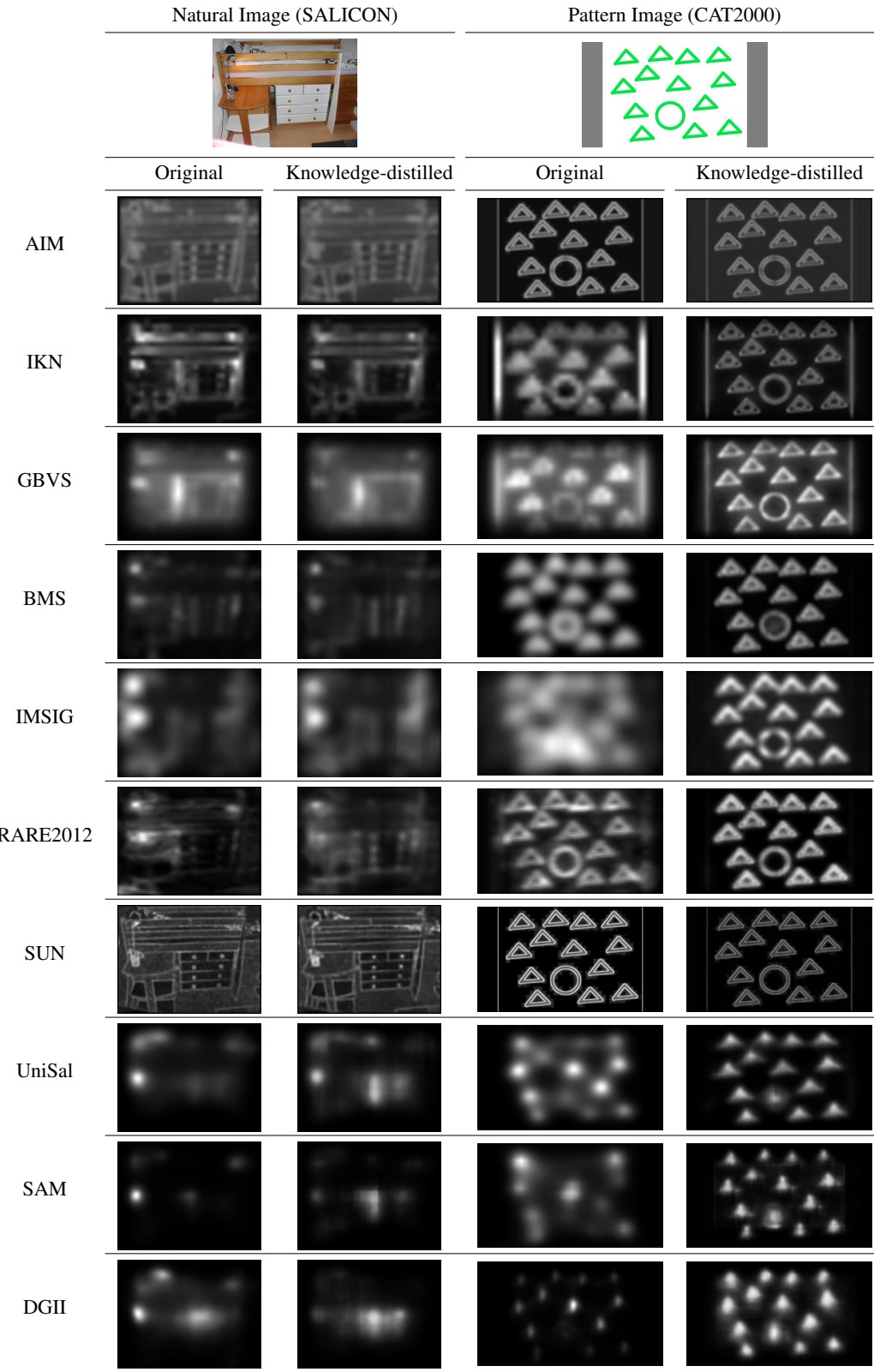

Figure 2: The performance of ten knowledged-distilled models in our KDSalBox shown for a sample image from both datasets: Natural Images (SALICON) and Patterns (CAT2000).

**Can the students generalize?**   Our student models were trained on saliency maps computed on the SALICON training images with the intention of capturing saliency behavior on a general sample of natural images. Therefore by design, our student models are intended to mimic the behavior of the original models on natural images, without guarantees about other image types. To investigate generalizability, in Table 2, we report the similarity of our student models to the original models on all 20 classes from the CAT2000 dataset by Borji & Itti (2015), which include art and pattern images, as well as on the 5 classes of the Imp1K dataset of graphic design classes Fosco et al. (2020). The knowledge-distilled AIM model generalized the best to other image types, performing least similar to the original AIM model on low resolution images in the CAT2000 dataset and infographics from Imp1K. The student models IKN, GBVS, BMS, IMSIG, and RARE2012 generalized moderately well, where the most difficult classes proved to be pattern, satelite, sketch, low resolution, and line drawing from the CAT2000 dataset, and infographics and mobile UIs from the Imp1K dataset. The three deep learning saliency models, UniSal, SAM, and DGII produced student models that generalized better on some classes than others. In Figure 2 we show how each model performs for two sample images, a natural image from the SALICON validation set, and a pattern image from the CAT2000 dataset. For the sample pattern image shown, all knowledge-distilled models output saliency along the edges of the triangles and the circle, whereas their original counterparts (especially the deep neural networks) predict differently, utilizing higher level features for the saliency maps. On the other hand, the original counterpart of the low level models (such as AIM, SUN, etc.) fail to capture these features as well, which explains why the knowledge-distilled models generalize well in this domain (i.e., reproducing their failures). In general, we caution against using the knowledge distilled models on images that deviate too much from the natural images they were trained on. Table 2 can be used to get an approximate sense of how representative a given model from the KDSalBox might be of the original saliency model on applications targeting specific image types.

**Data preprocessing during training.**   Models differ in the ranges of their saliency maps (e.g., compare AIM and SAM in Figure 2). It is difficult to visually compare the suitability of models for an application by ignoring these differences in saliency values, independently of the actual locations the models predict. For this reason we provide histogram matching as a possible processing step to facilitate apples-to-apples model comparisons. For these reasons, we also trained histogram matched model versions, but did not find that they improved model performance (more details in Appendix A).

## 4   Discussion

**Contributions.**   The toolbox presented in this work provides an approachable framework to conduct research on saliency with minimal effort. Our approach of offering a standardized but also very simple interface and environment for all saliency models can find use in the research community and in industry. Users of the toolbox can currently generate saliency images from ten saliency models with great efficiency, to evaluate them for application purposes. To allow the toolbox to continue to evolve, we also provide code to knowledge distill other models using the student architecture and training procedure we followed.

**Limitations and Future Work.**   In our work, we have used a unified architecture for all models for simplicity and easier maintainability. Depending on the application and setting (e.g. on-device deployment) a further reduction in the size or inference time might be needed in practice. This can be accomplished by other techniques such as optimizing for the memory hierarchy, for special instruction sets, change of datatypes, etc. Depending on the model, one can also design separate decoders that are more suitable for each model. This, however, comes at the expense of simplicity and generalizability. Another current limitation is that our models are trained only on the natural images from the SALICON dataset and as our preliminary generalization experiments show, some models (particularly students of deep neural net based teacher models) struggle more with generalizing to new image types than others. Extensions of this work call for training on other domains and with larger datasets. Finally, since all models share the same architecture and reside in the same framework, a natural extension of this work would be to combine these models like building blocks to increase performance on saliency benchmarks and on particular tasks where different models can offer complementary features (combined expertise).

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

Table 3: The ten models currently part of the KDSalBox trained on the histogram-matched images.

| Model | (Dis)similarity between student-teacher | | | |
|---|---|---|---|---|
| | Natural Images (SALICON) | | Patterns (CAT2000) | |
| | CC ↑ | KL ↓ | CC ↑ | KL ↓ |
| AIM | 0.98 ±0.01 | 0.04 ±0.10 | 0.87 ±0.10 | 3.03 ±3.00 |
| IKN | 0.97 ±0.02 | 0.04 ±0.03 | 0.73 ±0.10 | 0.29 ±0.12 |
| GBVS | 0.97 ±0.02 | 0.03 ±0.02 | 0.71 ±0.09 | 0.30 ±0.09 |
| BMS | 0.89 ±0.07 | 0.14 ±0.08 | 0.62 ±0.17 | 1.12 ±0.81 |
| IMSIG | 0.94 ±0.05 | 0.08 ±0.10 | 0.48 ±0.16 | 0.86 ±0.46 |
| RARE2012 | 0.86 ±0.11 | 0.19 ±0.12 | 0.63 ±0.14 | 0.74 ±0.50 |
| SUN | 0.93 ±0.06 | 0.09 ±0.08 | 0.82 ±0.11 | 2.71 ±2.11 |
| UniSal | 0.93 ±0.05 | 0.07 ±0.04 | 0.57 ±0.12 | 0.35 ±0.09 |
| SAM | 0.80 ±0.09 | 0.86 ±0.43 | 0.48 ±0.14 | 0.90 ±0.37 |
| DGII | 0.95 ±0.04 | 0.06 ±0.04 | 0.73 ±0.12 | 0.21 ±0.08 |

## A   Histogram Analysis

As in Table 1, we report our results in Table 3 for the case where the students are trained after all saliency maps are matched to the histogram of the ground truth of SALICON dataset. This makes it possible to compare the models with each other and facilitate the evaluation and selection of saliency models for different applications. As we can see the trained student models perform similarly as in the case without histogram matching in the SALICON dataset, but illustrate better where the saliency is distributed across the images. This is however not the case for the Pattern dataset, since the distribution in these images is quite different and reduces the ability of the student models to mimic their original counterparts. We also show in Figure 3 a sample image for both datasets for the histogram-matched case. Similar to our analysis in Section 3, the same conclusions apply here as well — i.e. the knowledge-distillation of the deep learning models generalize worse then the lower-level models. A potential way to improve this gap is to use a separate knowledge-distillation procedure for the deep learning models and use the intermediate layers to guide the training process (Romero et al. (2014)).

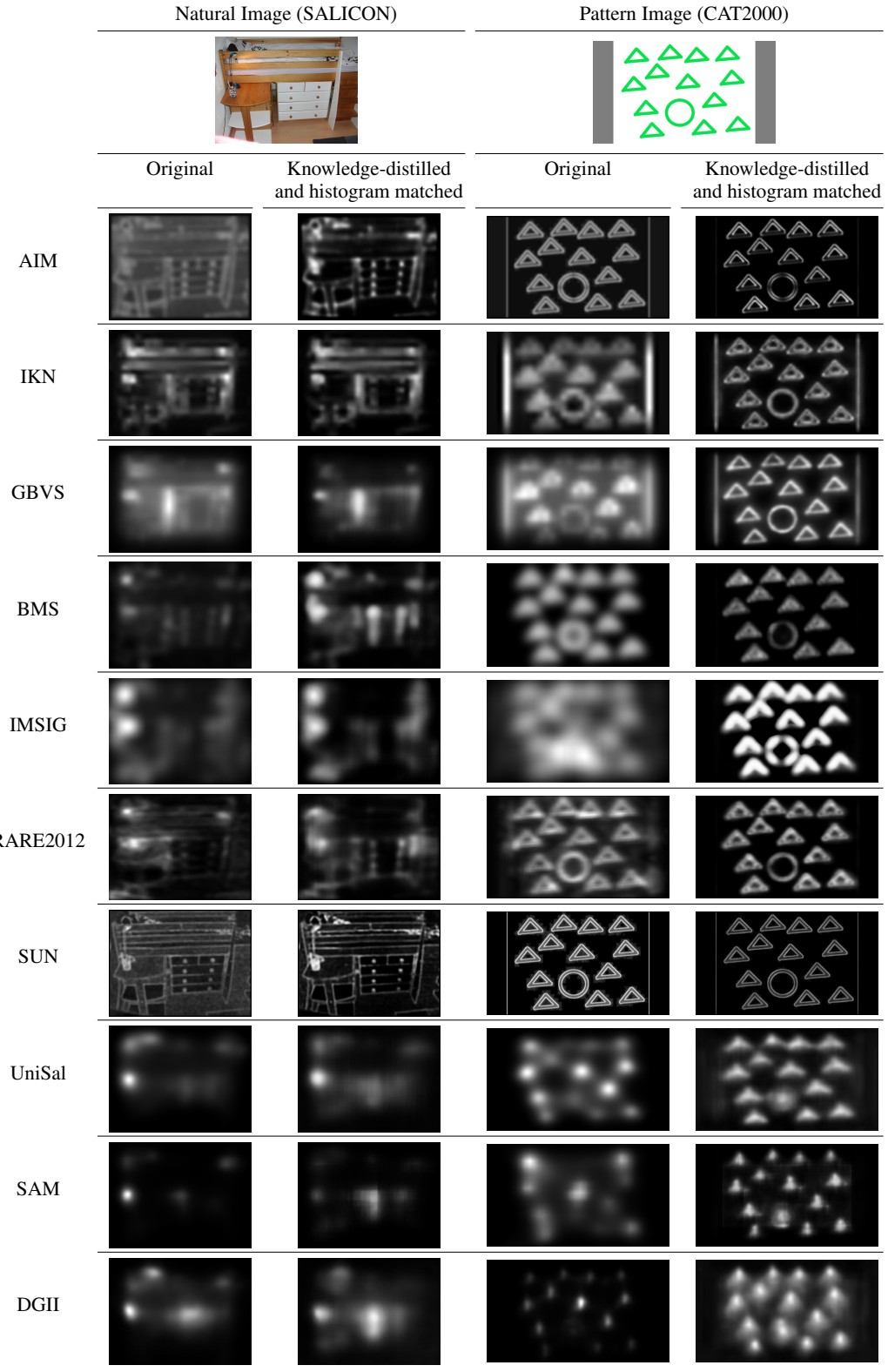

Figure 3: The performance of knowledged-distilled models after histogram-matching to the mean histogram of ground truth in the SALICON dataset shown for a sample image from both datasets: Natural Images (SALICON) and Patterns (CAT2000).

