# OpenReview forum: "KDSalBox: A toolbox of efficient knowledge-distilled saliency models"
_NeurIPS.cc/2021/Workshop/SVRHM — SVRHM 2021 Poster_

### Official Review · Reviewer_oqZH · 2021-10-27
**An engineering paper offering little scientific insight**

**Rating:** 3
**Confidence:** 4

**Review:**

This paper trains low-computational-cost MobileNet neural network architectures to mimic the two dimensional outputs of ten popular saliency models, and provides code for these knowledge-distilled models in a single package with a unified user interface. This contribution will be practically useful for members of the saliency community (the major point below notwithstanding). However, since I see little scientific contribution, I question whether SVRHM is the right venue for publication. Furthermore, I am worried about the potential for unfaithful model representations outside the training domain, which could cause unfair comparisions and evaluations of the models.

## Major

The paper notes that predictions for the knowledge-distilled networks are worse on non-natural images, particularly for the deep nets. I am worried that the mimicry performance of the distilled deep-net models may in fact be overestimated even within the class of natural images, because knowledge distillation is performed using the same dataset (SALICON) that many (all?) of these models are trained on. This may mean that the variance of the original model predictions is less on the SALICON set, compared to novel natural images (let alone the non-natural images compared in the paper), and that the distilled models learn this even-more-restricted set. One would have to carefully look at the training details of these models to decide whether the SALICON validation set can truly be considered a validation set for distillation (line 121). While the authors discusss limitations on generality (lines 162--163), I don't think these points go far enough: using the distilled models for comparison and evaluation on new image sets (even natural images) may not produce fair results compared to the original models.


## Minor

- The paper claims that low-level image feature models "fare better for certain applications", citing a paper from 2015 as evidence (line 23). Is this statement still true in the deep learning saliency era?
- line 41: Judd et al (2009) should also be cited when referring to the MIT Saliency Benchmark. In addition, the benchmark is now the "MIT/Tubingen Saliency Benchmark" (https://saliency.tuebingen.ai).
- Wrong authors for the GBVS paper. The authors are Harel, Koch and Perona (the authors listed in the references were the editors of NIPS that year).


## References

Judd, T., Ehinger, K. A., Durand, F., & Torralba, A. (2009). Learning to predict where humans look. IEEE 12th International Conference on Computer Vision, 2106–2113. https://doi.org/10.1109/ICCV.2009.5459462

---

> ### Author Response · Authors · 2021-12-10
> **Response to Reviewer oqZH**
>
> **Contribution**
>
> Regarding the scientific contribution of the paper, we make more explicit that we see this as “a technical report to supplement the release of our KDSalBox toolbox of saliency models” (pg. 2), which as the reviewer notes, should have a practical use for the saliency community.
>
> **Validation set of SALICON is not enough for generalization**
>
> We acknowledge the fact that the SALICON validation dataset can lead to overestimated results even for natural images. However, this dataset is a standard dataset used for the evaluation of saliency models and for drawing conclusions about their generalization in natural images. However, we agree that this limitation must be made explicit in our analysis. In addition, our OOD experiment results show that for Matlab-based models our knowledge-distilled models perform well even on the CAT2000 dataset, despite the varied natural image categories. As for the deep-learning-based models, as shown in the paper, they perform worse on OOD images, and as such their generalization is truly limited. We discuss this in greater detail in the paper and caution the users of the toolbox to consult Table 2 as an estimate of how the different knowledge-distilled models generalize to different image types. The main goal of this work is to facilitate the evaluation and selection of saliency models for different applications. The original models that are Matlab-based are not efficient and require a Matlab license to run. Our knowledge-distilled models can be beneficial in cases where the behavior of the original models is better and efficiency is important. The knowledge-distilled models are not meant to replace the original models for every case in an absolute sense. As they are knowledge-distilled versions they should be used always with their limitations in mind. Their main goal is to facilitate the use of the models for application purposes.

---

### Official Review · Reviewer_E4Cg · 2021-10-31
**This toolbox provides several older and modern saliency models by distilling the legacy software implementations into models based on MobileNet architectures. They demonstrate it works well on the SALICON dataset and briefly experiment with a synthetic image dataset: CAT2000.**

**Rating:** 7
**Confidence:** 4

**Review:**

While saliency methods have moved towards deep nets trained on the SALICON dataset, there are a number of legacy methods from the 1990s to now that are based on lower level image features and still interesting to study. However the software that implements them is aging, has difficult dependencies and may require commercial (e.g. Matlab) licences. This paper proposes to use SALICON to distill the knowledge of these legacy saliency models into MobileNet-based architectures.

Pros
- Really clever method; easy to use and maintain these MobileNet methods vs. the original salency model implementations
- Along with the distilled models, implements some extra features: a user interface and functions for distilling additional models
- It seems to work, providing a large speed advantage over the original models

Cons
- Though the OOD experiments are interesting, a more substantial realistic (compared to the synthetic dataset-based) OOD evaluation would improve the paper

The key idea leveraged by this paper, that is, distilling legacy software implementations is really clever and generally applicable beyond saliency. I wonder if the authors saw this idea elsewhere or it was born here in saliency models? I would appreciate some discussion on this point. I'm curious if it has been applied much elsewhere. I am familiar with knowledge distillation models but haven't seen this legacy software application.

One question I had was that the encoder used a standard architecture: MobileNetV2 but the decoder seemed rather arbitrary ("Through some pilot testing, we converged on a decoder that was simple and compact". Was it not possible to use some standard decoder from e.g. semantic segmentation to keep the architectural elements standard?

The point about the Matlab-based models being easier to reproduce than the DL-based models is really interesting. The authors hypothesize that it is due to the limited dataset size during training, making it "difficult to capture high-level features, as well as the simple model architecture we used". Could it also be that the Matlab-based models use simpler (hand-designed) functions that are just easier to learn with a parameter-restricted net?

Typos
- Model architecture paragraph: "performing well i knowledge distilling"
- Contributions paragraph in Discussion: "and in industry" (remove "the")

---

> ### Author Response · Authors · 2021-12-10
> **Response to Reviewer E4Cg**
>
> **Though the OOD experiments are interesting, a more substantial realistic (compared to the synthetic dataset-based) OOD evaluation would improve the paper**
>
> We have extended our experiments with further OOD experiments. We have now included in the paper the evaluation scores for all 20 image categories from CAT2000 as well as 5 categories of graphic design images from the Imp1k dataset. The new experiments confirm and strengthen further the conclusions that we derived with our initial experiments.
>
> **Decoder seemed rather arbitrary**
>
> Our decoder is most similar to the standard decoders in semantic segmentation, and it uses common building blocks including upsampling and convolutional layers. Our goal was to find a simple decoder that could perform well for all our saliency models, and our tests culminated in a decoder with 6 convolutional layers and 5 intermediate bilinear upsampling layers, motivated by the fact that the original MobileNetv2 downsamples the original image (h x w) to a size (h//32 x w//32).
>
> **Matlab-based models use simpler (hand-designed) functions that are just easier to learn with a parameter-restricted net**
>
> We agree that the main reason why the Matlab-based models generalize better is that they are more simple and capture only low-level features, which subsequently is easier to be learned by our knowledge-distilled models.

---

### Official Review · Reviewer_J1RP · 2021-10-31
**Very interesting and promising work**

**Rating:** 7
**Confidence:** 4

**Review:**

The authors present a method to build knowledge-distilled saliency models from previously published models, and they aim to release the toolbox as a user-friendly platform.

The methodology is correct, and I only suggest better explaining why only CC and KL metrics are reported, and not NSS or AUC.

I understand that for the general public an easy-to-use platform with several available models is useful. But I wonder if this should be recommended, as the authors show that the students differ more from the teachers in the more recent models than in older models. And conversely, the more recent models have better overall performance in the datasets presented here and are also available in Python, while the older ones have poorer performance and older codes in Matlab (not open-source). Why is not better to put the effort in code the older models in Python when needed and make the code available? What else is possible to learn from the proposed approach? I suggest adding a discussion on this if the manuscript is accepted.

Minor comments.
* There is a typo in line 102: “... performing well i knowledge…”

---

> ### Author Response · Authors · 2021-12-10
> **Response to Reviewer J1RP**
>
> **Better explaining why only CC and KL metrics are reported, and not NSS or AUC**
>
> During the evaluation, we compare saliency maps produced by the student model compared to the teacher model. Since we are comparing distributions, the use of distribution-based metrics CC and KL is most appropriate. NSS and AUC are most frequently used to evaluate the ability of a saliency map (distribution) to predict ground truth fixation locations (binary mask with 1 wherever there is a fixation). While there are ways to adapt these metrics to comparing distributions, this is not standard practice for saliency evaluation.
>
> **Students differ more from the teachers in the more recent models**
>
> We acknowledge the limitation of our approach for the recent models and the difficulties in creating knowledge-distilled models that perform well in approximating more recent (deep-learning-based) models. This holds true, especially when generalizing to other image types such as the ones in CAT2000. A more complex decoder or training procedure (such as using the features in intermediate layers) is a direction of future work that we are considering.
>
> **Why is not better to put the effort in code the older models in Python when needed and make the code available? What else is possible to learn from the proposed approach?**
>
> Besides having a unified framework for all the models, the method of using knowledge-distilled models instead of the original models also has its own value, since the models are more efficient. This approach makes them usable in applications where the original model is impractical. For example, the SUN model is very inefficient and requires several seconds to be run (no matter whether it is implemented in Matlab or Python), whereas its counterpart requires half a second on a CPU and 0.01 seconds on a GPU. Finally, as discussed in Section 4, having all models implemented in a common framework opens up opportunities for combining models like building blocks for new applications.

---

### Decision · Program_Chairs · 2021-11-02

Accept (Poster)